# Early Rehabilitation Exercise after Stroke Improves Neurological Recovery through Enhancing Angiogenesis in Patients and Cerebral Ischemia Rat Model

**DOI:** 10.3390/ijms231810508

**Published:** 2022-09-10

**Authors:** Huixia Geng, Min Li, Jing Tang, Qing Lv, Ruiling Li, Lai Wang

**Affiliations:** 1Institute of Chronic Disease Risks Assessment, School of Nursing and Health Sciences, Henan University, Kaifeng 475004, China; 2The School of Life Sciences, Henan University, Kaifeng 475000, China

**Keywords:** stroke, early rehabilitation, neurological function, angiogenesis

## Abstract

Among cerebrovascular diseases, ischemic stroke is a leading cause of mortality and disability. Thrombolytic therapy with tissue plasminogen activator is the first choice for clinical treatment, but its use is limited due to the high requirements of patient characteristics. Therefore, the choice of neurological rehabilitation strategies after stroke is an important prevention and treatment strategy to promote the recovery of neurological function in patients. This study shows that rehabilitation exercise 24 h after stroke can significantly improve the neurological function (6.47 ± 1.589 vs. 3.21 ± 1.069 and 0.76 ± 0.852), exercise ability (15.68 ± 5.95 vs. 162.32 ± 9.286 and 91.18 ± 7.377), daily living ability (23.37 ± 5.196 vs. 66.95 ± 4.707 and 6.55 ± 2.873), and quality of life (114.39 ± 7.772 vs. 168.61 ± 6.323 and 215.95 ± 10.977) of patients after 1 month and 3 months, and its ability to promote rehabilitation is better than that of rehabilitation exercise administered to patients 72 h after stroke (*p* < 0.001). Animal experiments show that treadmill exercise 24 h after middle cerebral artery occlusion and reperfusion can inhibit neuronal apoptosis, reduce the volume of cerebral infarction on the third (15.04 ± 1.07% vs. 30.67 ± 3.06%) and fifth (8.33 ± 1.53% vs. 30.67 ± 3.06%) days, and promote the recovery of neurological function on the third (7.22 ± 1.478 vs. 8.28 ± 1.018) and fifth (4.44 ± 0.784 vs. 6.00 ± 0.767) days. Mechanistic studies have shown that treadmill exercise increases the density of microvessels, regulates angiogenesis, and promotes the recovery of nerve function by upregulating the expression of vascular endothelial growth factor and laminin. This study shows that rehabilitation exercise 24 h after stroke is conducive to promoting the recovery of patients’ neurological function, and provides a scientific reference for the clinical rehabilitation of stroke patients.

## 1. Introduction

Stroke has the characteristics of high morbidity, high disability rate, high mortality, and heavy disease burden at a global level. Cerebral ischemic stroke (CIS) accounts for 80% of all stroke patients, is a common type of clinical stroke, and is also the leading cerebrovascular accident leading to patient death and disability [1]. After stroke, the interruption of cerebral blood flow causes a lack of blood supply to the distal brain tissue, which eventually leads to cell apoptosis, resulting in neurological injury in patients. Removal of thrombi from cerebral blood vessels by tissue plasminogen activator (t-PA) or mechanical thrombectomy is the main treatment to restore blood flow in patients after stroke. However, re-establishment of blood flow by dissolving blood clots paradoxically leads to delayed neuronal apoptosis itself. Therefore, reducing neuronal apoptosis and promoting neurological recovery after cerebral ischemia-reperfusion are important therapeutic strategies for stroke patients.

Stroke is one of the major causes of long-term disability, and effective post-stroke rehabilitation is beneficial to the recovery of neurological function in moderate or severe stroke victims [2]. Constraint-induced movement therapy (CIMT) is widely used for the rehabilitation of motor function in post-stroke patients, but there are some shortcomings, such as inconsistent immobilization durations and methods, narrow application scope, and an emotional impact [3,4]. Mindfulness-based stress reduction (MBSR) was created by Dr. Jon Kabat-Zinn in 1979 and promotes function across a wide range of chronic disorders and diseases by involving training in mindful meditation [5,6,7]. In recent years, MBSR has been used to promote stroke recovery, especially for psychological and cognitive function [8,9]. The Bobath approach, also known as the neuro-developmental technique (NDT), is widely used in the rehabilitation of stroke patients, but the effectiveness of this rehabilitation method is controversial [10,11].

Evidence-based early intervention of rehabilitation measures can improve the daily self-care ability of post-stroke patients, effectively reduce the neurological dysfunction of patients, and significantly improve the quality of life, thereby reducing potential medical care costs and saving social resources [12]. According to the timepoint of the rehabilitation training intervention, rehabilitation after stroke is divided into: ultra-early rehabilitation (≤24 h), early rehabilitation (1–7 day), and subacute rehabilitation (7 days–3 month) [13,14,15]. Rehabilitation training as early as possible after stroke patients reach a relatively steady state can promote the recovery of patients’ neurological function and improve the quality of life of patients. The concept of early rehabilitation after stroke has been recognized internationally, but the types and methods of rehabilitation training are still controversial, especially the choice of rehabilitation time [16,17]. Some studies have pointed out that early rehabilitation may lead to risks such as aggravation of neurological deficits, and some studies have pointed out that early rehabilitation can speed up the restoration of neurological function, and effectively reduce complications [16,17,18]. Although it has been widely accepted that the sooner the rehabilitation training starts after a stroke, the better the recovery of the patient’s neurological function, none of the guidelines provide exact guidance on the specific intervention time of early rehabilitation [19].

In this study, stroke patients were selected as the research objects, and two timepoints, 24 h and 72 h after stroke, were selected as the start times for rehabilitation training. The effects of early rehabilitation training at different timepoints on the neurological function of patients with acute ischemic stroke were compared. In addition, this study took the middle cerebral artery occlusion (MCAO) rat model as the research object to explore the molecular mechanism of early rehabilitation training to promote neurological recovery. The purpose of this study is to provide a scientific, targeted, and rational basis for the selection of early rehabilitation training timing for patients with acute ischemic stroke.

## 2. Results

### 2.1. The Effects of Different Rehabilitation Training Timings on the Neurological Function and Motor Ability of the Patients before and after Rehabilitation

At 1 month and 3 months after rehabilitation exercise, the neurological functions of the two groups were significantly improved when compared with those before rehabilitation. The NIHSS score of the 24 h rehabilitation group and the 72 h rehabilitation group decreased from 3.21 ± 1.069 and 5.05 ± 1.246 to 0.76 ± 0.852 and 2.51 ± 0.989, respectively, and the score of the 24 h rehabilitation group was significantly lower than that of the 72 h rehabilitation group (*p* < 0.05) (Table 1). The FMA score showed that motor function after rehabilitation was significantly improved in comparison to that before rehabilitation. The FMA score of the 24 h rehabilitation group was 62.32 ± 9.286 and 91.18 ± 7.377 at 1 month and 3 months after rehabilitation, and the results were significantly better than that of the 72 h rehabilitation group (41.43 ± 7.065 and 73.22 ± 7.811). The statistical analysis showed significant difference (*p* < 0.05) (Table 2).

### 2.2. Effects of Different Rehabilitation Training Timings on ADL Ability and Quality of Life of the Patients before and after Rehabilitation

Before rehabilitation, the ADL scores of patients in the 24 h and 72 h rehabilitation groups were 23.37 ± 5.196 and 22.84 ± 5.036, respectively, with no significant difference (*p* > 0.05). At 1 month and 3 months of rehabilitation, the ADL scores of the patients in the 24 h rehabilitation group were 66.95 ± 4.707 and 96.55 ± 2.873, respectively, which were higher than those in the 72 h rehabilitation group (52.95 ± 6.725 and 84.46 ± 6.971), and the difference was extremely significant (*p* < 0.001) (Table 3). The quality of life scores showed that after 1 month and 3 months of rehabilitation exercise, the quality of life scores of the two groups of patients were improved in comparison to those before rehabilitation. Compared between groups, the quality of daily life of patients in the 24 h rehabilitation group was better than that of the 72 h rehabilitation group at 1 month and 3 months after rehabilitation exercise (*p* < 0.001) (Table 4).

### 2.3. The Effect of Treadmill Exercise on Cerebral Infarction Volume and Neurological Function 

TTC staining was used to detect the cerebral infarction volume in each group after middle cerebral artery occlusion. The results showed that the volume of cerebral infarction of MCAO group is 30.67 ± 3.06%, and with the progress of rehabilitation training, the volume of cerebral infarction gradually decreased on the third (15.04 ± 1.07% vs. 30.67 ± 3.06%, *p* < 0.01) and fifth (8.33 ± 1.53% vs. 30.67 ± 3.06%, *p* < 0.01) days in the 24 h treadmill exercise group, and the difference was statistically significant. This shows that rehabilitation training at 24 h after MCAO can reduce the volume of cerebral infarction and reduce brain damage caused by cerebral ischemia-reperfusion (Figure 1).

The results of the neurological function analysis showed that the mNSS score of the rats in the 24 h rehabilitation training group was lower than that of the MCAO group on the third (7.22 ± 1.478 vs. 8.28 ± 1.018) and fifth (4.44 ± 0.784 vs. 6.00 ± 0.767) days, and the difference was statistically significant (*p* < 0.05). This shows that rehabilitation training 24 h after cerebral ischemia-reperfusion can effectively promote the recovery of neurological function in rats (Table 5).

### 2.4. The Effect of Treadmill Exercise on Neuronal Apoptosis after MCAO and Treadmill Exercise 

Western blot was used to detect the expression changes of apoptosis-regulating proteins in the brain of each group. The results showed that rehabilitation training reduced the expression of the activated forms of caspase 3 and caspase 8, and the decreasing trend became more significant with the increase in training time, suggesting that rehabilitation training starting at 24 h reduced neuronal apoptosis and enhanced neuronal survival (Figure 2).

### 2.5. The Effect of Treadmill Exercise on Angiogenesis in the Ischemic Penumbra

After cerebral ischemia-reperfusion, reducing neuronal apoptosis is an important method of promoting the recovery of neurological function. However, promoting angiogenesis is an important factor for maintaining neuronal survival. The expression of vascular endothelial growth factor (VEGF) in the rat brain on day 1, day 3, and day 5 in the 24 h treadmill exercise group after MCAO was significantly increased when compared to the MCAO group, and the differences were statistical significance (Figure 3). The expression level of laminin is consistent with the expression of VEGF. Treadmill rehabilitation training in the 24 h treadmill exercise group after MCAO upregulates the expression of laminin and increases the density of microvessels, suggesting that exercise rehabilitation can enhance angiogenesis in the brain ischemic penumbra. This suggests that treadmill exercise can enhance angiogenesis and promote the recovery of neurological function (Figure 4).

## 3. Discussion

Stroke is currently a major cause of disability and death for adults in the world [20]. The pathological injury mechanism of stroke is complex, and there is no effective drug for clinical treatment. Thrombolytic therapy after acute cerebral ischemia has limited use due to its various limitations. Even after timely treatment of ischemic stroke, different types of dysfunction are often left behind. The neurological and motor functions of patients are damaged, which leads to an inability to return to family and social life normally, which seriously affects their quality of life [21]. Clinical statistics show that about 15–30% of stroke survivors will become permanently disabled [22]. Therefore, exploring strategies and methods for neurological recovery of patients after a stroke is crucial to improving the quality of life of patients.

Substantial progress has been made in the research on post-stroke rehabilitation, and targeted motor function recovery can promote the motor function of ischemic stroke patients [23,24,25,26]. However, the results of animal experiments and clinical studies have shown that the choice of rehabilitation timing after stroke is controversial. A large number of animal studies have shown that exercise training within 24–48 h after stroke reduces neuroinflammation, maintains blood–brain barrier integrity, increases BDNF expression, inhibits neuronal apoptosis, and promotes nerve regeneration, which has better neurological behavior and smaller cerebral infarction volume [27,28,29,30]. However, some studies have also shown that rats that underwent treadmill exercise 24 h after MCAO have increased infarct volume of cerebral ischemia, worse behavioral outcomes, and less neuronal proliferation compared with control rats [31,32]. Studies in human trials (a very early rehabilitation trial for stroke) have shown that early exercise rehabilitation within 24 h after stroke is potentially harmful, and other studies have shown beneficial effects of rehabilitation training at the same timepoint [33]. A recovery time from 24 h to 3 months was selected; however, the timepoint that is most conducive to the recovery of neurological function of patients is still controversial [34].

The results of this study showed that after 1 month of rehabilitation training, the NIHSS score of the rehabilitation training group at 24 h post-stroke was 3.21 ± 1.069 points, and the NIHSS score of the 72 h rehabilitation group was 5.05 ± 1.246 points. After 3 months of rehabilitation, the score of the 24 h rehabilitation group was 0.76 ± 0.852 points, and the score of the 72 h rehabilitation group was 2.51 ± 0.989 points. With the intervention and progress of rehabilitation training, the NIHSS scores of both the 24 h rehabilitation group and the 72 h rehabilitation group were significantly lower than those before rehabilitation, the patient’s neurological function was subsequently significantly improved, and the difference was significant (*p* < 0.05). This suggests that rehabilitation training has a positive effect on the recovery of the nervous system function of acute stroke patients.

The results of this study also showed that the NIHSS scores of the two groups of patients were different. After 1 month of rehabilitation, the neurological function of the 24 h rehabilitation group was better than that of the 72 h rehabilitation group and after 3 months of rehabilitation, the neurological function of the 24 h rehabilitation group was also better than the 72 h rehabilitation group. The results suggest that 24 h early rehabilitation training can promote the degree and speed of neurological recovery in patients, resulting in more favorable rehabilitation outcomes for patients. The reason may be that after cerebral ischemia, brain neurons and brain tissue still have a certain degree of plasticity. Through rehabilitation training, the establishment of collateral circulation in the brain injury area can be accelerated, the degree of brain ischemia injury can be reduced, and the formation of new synapses can be promoted in order to achieve neurological recovery. Bernhardt’s study also showed that rehabilitation within 24 h after stroke is safe and does not increase the mortality rate of patients within 3 months [35,36].

The Fugl-Meyer Assessment (FMA) scale is an important index for assessing motor function, sensation, balance, joint range of motion, and joint pain in stroke patients [37]. In this study, the FMA was used to evaluate the effect of rehabilitation exercise on the recovery of limb motor function in patients with acute ischemic stroke at 24 h and 72 h after acute ischemic stroke. The results showed that the FMA scores of the two groups of patients after 1 month and 3 months of rehabilitation were significantly higher than those before rehabilitation, and the difference was significant (*p* < 0.05). This shows that the patient’s motor function has been effectively improved with the progress of rehabilitation training. There were significant differences in the FMA scores between the two groups at 1 month of rehabilitation training and 3 months of rehabilitation training (*p* < 0.05). At 1 month and 3 months of rehabilitation training, the FMA scores of the patients in the 24 h rehabilitation group were significantly better than those in the 72 h rehabilitation group, suggesting that the recovery of motor function of the patients in the 24 h rehabilitation group was faster and better than that of the 72 h rehabilitation group. 

In the rehabilitation process of stroke patients, the improvement of walking ability is the most frequently concerned aspect by stroke patients and their families. The ability to walk independently is often regarded as one of the influencing factors of patients’ willingness to be discharged from the hospital, and it is also a crucial factor for doctors to judge whether stroke patients can be discharged from the hospital. Gumming et al. pointed out that early rehabilitation training can significantly shorten the number of days for stroke patients to recover their ability to walk independently and accelerate the process of patients’ motor function recovery through a clinical randomized controlled trail. Early rehabilitation training is a fast path for patients with acute stroke to restore independent walking ability, which can directly promote the recovery and discharge of patients and return to life and society [16]. Some studies have also pointed out that exercise rehabilitation activities for acute stroke patients within 24 h after stroke have a significant effect on the recovery of upper limb function of patients [38].

Activities of daily living (ADL) refer to the most basic and most common activities that individuals repeat every day in order to maintain their survival and meet their daily needs, that is, a series of activities necessary for an individual in terms of clothing, walking, eating, housing, transportation, maintaining personal hygiene, and conducting independent social activities [39]. The ability to perform the activities of daily living is an important indicator that reflects the health of stroke patients. It can determine whether the patient has the ability to live independently. It is an important part of the rehabilitation of stroke patients. In the results of this study, the ADL scores of patients in the 24 h rehabilitation group were significantly higher than those in the 72 h rehabilitation group at 1 and 3 months follow rehabilitation treatment, suggesting that rehabilitation training 24 h after stroke has a better effect on the recovery of patients’ ADL. Some studies believe that the first 3 months after stroke have the fastest recovery speed, which is crucial to the recovery of patients, emphasizing the necessity of early rehabilitation in rehabilitation training, so that the self-care ability of stroke patients can be recovered earlier [40]. The study by Cumming et al. showed that the patients who got out of bed within 24 h after stroke had much higher ADL scores than that of the delayed activity group after 3 months of follow up, and after multiple regression analysis, they pointed out that the patients in the ultra-early activity group had better motor function and independence in activities of daily living [41].

The results of the quality of life survey showed that the quality of life scores of the patients in the 24 h rehabilitation group were higher than those in the 72 h rehabilitation group at 1 month and 3 months of rehabilitation exercise, suggesting that early rehabilitation training had a positive effect on the improvement of patients’ quality of life, and that rehabilitation training 24 h after stroke is more conducive to the improvement of patients’ quality of life. These effects may be related to the fact that early rehabilitation training helps to promote the recovery of neurological and motor functions and enables patients to better adapt to autonomous life, thereby improving the quality of life of patients.

The results of animal experiments in this study showed that the treadmill exercise 24 h after middle cerebral artery occlusion and reperfusion reduced the volume of cerebral infarction in rats, inhibited cell apoptosis, decreased neurological function scores, and enhanced neurological function recovery. This study used the rat MCAO model to investigate the mechanism by which rehabilitation exercise within 24 h after stroke promotes neurological function. A large number of studies have shown that aerobic exercise is beneficial to promote the recovery of neurological function after stroke. Treadmill high-intensity interval training for 20 min a day elicited significantly greater acute increases the level of BDNF in the serum of post-stroke patients and corticospinal excitability [42]. Vascular endothelial growth factor (VEGF) promotes angiogenesis and maintains vascular homeostasis by promoting the proliferation, survival, and migration of endothelial cells [43]. In general, angiogenesis is observed in the ischemic penumbra within 4 to 7 days after cerebral ischemia, and VEGF expressed by endothelial cells is conducive to angiogenesis and promotes the recovery of neurological function after cerebral ischemia [44]. The results of animal experiments in this study showed that treadmill exercise 24 h after middle cerebral artery occlusion and reperfusion increased the expression of VEGF in the ischemic penumbra, suggesting that early rehabilitation exercise may reduce neuronal apoptosis and promote neurological recovery by promoting angiogenesis.

Laminin is a glycoprotein, which is an important structural component in the extracellular matrix, has multiple functions in the central nervous system, and is widely involved in the development of the nervous system [45]. In addition, laminin is a key molecular constituent of the vascular basal lamina; therefore, it has been extensively used as an excellent marker for the density of the microvascular network [46]. Studies have reported that laminin can repair the disruption of the blood–brain barrier and the remodeling of the endothelial barrier by promoting new angiogenesis [47]. Elevated expression of laminin is beneficial to wound healing and angiogenesis [48]. This study shows that after middle cerebral artery occlusion and reperfusion, the expression level of laminin decreases, and treadmill exercise after 24 h promotes the increase in laminin expression, indicating that rehabilitation exercise after 24 h promotes angiogenesis, which is consistent with the high expression of laminin and neurological recovery [49].

Although clinical guidelines in recent years have pointed out that rehabilitation training should be as soon as possible post-stroke, the optimal time for rehabilitation is still uncertain. This study emphasizes gradual progress in the rehabilitation program. The results suggest that rehabilitation training at 24 h and 72 h after stroke can effectively improve the patient’s neurological and motor functions, and enhance the patient’s ability to live independently, improving their quality of life. The results of this study also suggest that 24 h rehabilitation training after stroke is more effective than 72 h rehabilitation training in the recovery of various functions in patients with acute ischemic stroke (*p* < 0.05). Animal experimental studies have shown that treadmill exercise 24 h after middle cerebral artery occlusion can promote the recovery of neurological function by upregulating the expression of VEGF, increasing the formation of blood vessels, reducing the apoptosis of neurons, reducing the volume of cerebral infarction, and reducing brain tissue damage. To summarize, rehabilitation training begun 24 h after stroke has a better prognosis for patients with acute ischemic stroke, the time required for patients to achieve functional recovery is shorter, and the effect is better, which can shorten the hospitalization time of patients. Patients and their families save medical expenses and improve social and economic benefits. However, this study also has some shortcomings, such as a small sample size and a short study time. In addition, the patient-specific physiology, including the intactness of circle of Willis and leptomeningeal collateral flow, has important guiding significance for evaluating cerebral perfusion and neurorehabilitation in stroke patients [50,51]. However, at the time of this study, these specific physiology of patients were not detected and evaluated. Therefore, it is necessary that patient-specific physiology is tested in future research on large-scale rehabilitation training clinical trials and research.

## 4. Materials and Methods

### 4.1. Patients

A total of 75 patients with acute ischemic stroke who were hospitalized in the department of neurology of a Level-3/Grade-A hospital in Kaifeng City, Henan Province from September 2019 to August 2020 were selected as the research subjects. Inclusion criteria: ① meet the diagnostic criteria revised in “ Chinese guidelines for diagnosis and treatment of acute ischemic stroke 2018” [3]; ② age ≥ 18 years; ③ admission to hospital within 24 h after stroke onset; ④ NIHSS score >1 and <22, with symptom stability; ⑤ signing of the informed consent form and participate voluntarily. Exclusion criteria: ① hemiplegia caused by other reasons; ② a history of mental disorders, combined with primary diseases of the heart, liver, and other systems, accompanied by infectious diseases such as viral hepatitis and tuberculosis; ③ severe joint deformity or joint disease before stroke, which would affect functional recovery. Termination criteria: ① new lesions or other diseases or accidents that seriously affect the rehabilitation and functional evaluation during the rehabilitation period; ② the patient or his family refused to continue the rehabilitation; ③ The patient obviously violated the rehabilitation programs. This study was approved by the Committee of Medical Ethics and Welfare for Experimental Animals, Henan University School of Medicine.

The sample size required for this study was estimated according to the sample size formula.
$$n=\frac{2{\left(\mu_{\alpha }+\mu_{\beta }\right)}^{2}\sigma^{2}}{\sigma^{2}}$$

In the formula, *n* is the number of samples required for each of the two groups and µα is the µ value corresponding to the probability of type I error. The corresponding values are looked up in a table, and can be found that µ0.05 = 1.96 for both sides. μβ is the μ value corresponding to the probability β of type II error, μ0.02 = 0.84. The total sample size with this formula was 68. Accounting for a 10% loss to follow-up rate and sampling error, the sample size was expanded to 75 cases.

The patients who met the inclusion criteria were randomly divided into the 24 h rehabilitation group with 38 cases and the 72 h rehabilitation group with 37 cases using the random number table method (Figure 5). The gender, age, hypertension, diabetes, cerebral infarction area, and other general data of the two groups of patients had no significant differences (*p* > 0.05). Basic information on the included patients is summarized in Appendix A. Both groups of patients received routine neurological treatment, nursing, and the same rehabilitation training program. Patients in the 24 h rehabilitation group started rehabilitation training at 24 h after stroke, and patients in the 72 h rehabilitation group received rehabilitation training at 72 h after stroke. After assessing the patient’s condition by a neurologist and a professional rehabilitation physician, the patient’s rehabilitation plan was jointly formulated by the professional team. During the actual process, the corresponding rehabilitation measures were given according to the patients’ condition within the scope of the plan, so that the patient did not feel fatigued. During the rehabilitation training, a nurse was required to be present, and the patient’s heart rate, blood pressure, pulse rate, respiration rate and other vital signs was closely monitored, and the patient was dynamically assessed.

### 4.2. Patient Rehabilitation

If the muscle strength was less than or equal to grade 1, the patients were instructed to perform supine position training, including correct placement of good limbs, maintaining anti-spasmodic position, position replacement, bridge exercise, etc. Professional rehabilitation specialists intervened to perform passive movement of limbs and joints for patients. The rehabilitation program was as follows: 30 min/time, 2 times/d, 6 d/week.

Grade 1 < muscle strength ≤ grade 3, on the basis of the previous stage, the activities range of joint were changed from large to small, and the amount and frequency of activities were gradually increased. Rehabilitation specialists instructed patients to perform exercises such as stretching hands and elbows, simulating face washing, bending and stretching knee and hip joints, and moving ankle and toe joints (5 times/time, 5 times/d). Rehabilitation specialists assisted patients in transitional training from supine to sitting position, training of back muscles, abdominal muscles and respiratory muscles, gradually adding shoulder and foot activities, Bobath handshake training, reflex anti-spasm training, sitting, standing balance training, etc.

Muscle strength > 3, with the help of rehabilitation specialists, the rehabilitation training gradually increased to bedside standing exercises, standing balance exercises, and the activity time was gradually extended. According to the state of the patient, the training of sit-ups, active joint training, walking training, fine function training, daily living ability training and occupational therapy, etc., were gradually increased. The rehabilitation program was as follows: 60 min/time, 2 times/d, 6 d/week.

### 4.3. Patient Limb Function Testing

The recovery effect of the patients was evaluated before rehabilitation, 1 month after rehabilitation, and 3 months after rehabilitation. ① Neurological function: The NIHSS was used to assess the neurological status, with a total of 42 points, with 0 point representing completely normal, higher scores indicate more severe neurological deficits in patients [52]. ② Motor function: The Fugl-Meyer Assessment (FMA) was selected to assess the motor function of patients. Each item has a three-level scoring system, corresponding to 0 to 2 points respectively, for a total of 100 points. A score of <50 indicates the presence of severe dyskinesia, and the lower the score, the worse the patient’s motor function [53]. ③ Activities of daily living (ADL) ability: The simplified Barthel scale (BI) was used to measure the daily living ability of patients, with a total of 100 points. The lower the score, the worse the daily living ability of the patient, and lower than 60 points indicates that there is a certain degree of self-care disorder [54]. ④ Quality of life: the stroke specialized quality of life scale (SS-QOL) is a special scale to test the quality of life of patients in the form of questions and answers, including 49 items in 12 domains such as mobility (6 items), energy (3 items), upper extremity function (5 items), work/productivity (3 items), mood (5 items), etc. The higher the score, the higher the level of quality of life of the patient [55].

### 4.4. Animals

Male adult Sprague–Dawley rats (SD), weighing 230–250 g, were purchased from Beijing Vital River Laboratory Animal Technology Co., Ltd. (Beijing, China) Experiments were carried out after a week of acclimatization in a specific pathogen-free laboratory with controlled temperature (20–25 °C) and humidity (50–60%). Both pelleted chow and water were sterilized and freely eaten and drank by rats. The rats were coded uniformly, and the numbers were randomly selected into the sham group, the MCAO group, and the 24 h rehabilitation training group. The samples were taken at the 1 d, 3 d, and 5 d timepoints after rehabilitation training. All experiments were approved by the Committee of Medical Ethics and Welfare for Experimental Animals, Henan University School of Medicine.

### 4.5. Middle Cerebral Artery Occlusion (MCAO)

Rats were fasted 12 h before surgery. The rats were anesthetized with 1% sodium pentobarbital by intraperitoneal anesthesia. During the operation, the rats were kept breathing spontaneously, and a constant temperature heating pad was used to keep the body temperature of the rats constant (37 ± 0.5 °C). After the rats were anesthetized, the neck fur was shaved, and the surgical site was disinfected by 70% ethanol. An incision was made in the middle of the neck with surgical scissors, and the muscle layer was separated under a dissecting microscope to expose the common carotid artery (CCA), external carotid artery (ECA), and internal carotid artery (ICA). A specific suture was inserted from the external carotid artery according to Longa’s classic method, which is detailed as follows: Tie a dead knot in the external carotid artery to close the blood vessel, tie a slip knot to fix the suture, use microscissors to cut a V-shaped incision, insert the suture, and fix by the slip knot. Untie the knot at the internal carotid artery, turn the suture so that the uppermost end of the suture is in the same direction as the internal carotid artery and slowly push the suture along the upper left until a slight resistance is felt. When the blood flow curve of the middle cerebral artery is observed to decrease by about 70–80% by laser doppler flowmeter, the advancing of the suture can be stopped, and the slip knot of the fixed suture can be lightly tied at the same time. After 2 h, the suture was removed and the slip knot on the common carotid artery was opened, which was reperfusion. The operation of the sham group was exactly the same as that of the model group except that the suture was inserted without blocking the blood flow of the middle cerebral artery.

### 4.6. Treadmill Exercise

Before the rehabilitation program, all rats were given adaptive treadmill exercise for 3 days on treadmill platform. The treadmill speed was set at 15 m/min and the time was set at 30 min/d. All three groups of rats were grasped under the same conditions. After MCAO, the model group and the sham group were put back into their original cages without any treatment and nursing measures, and they were allowed to eat and drink freely. The rehabilitation group were started treadmill training at 24 h after MCAO, and the rehabilitation program was determined as the one formed in the adaptive training programs. The rat brains were sampled on days 1, 3, and 5 after treadmill training for further analysis.

### 4.7. Infarct Volume Analyses

After the treadmill training, the rats were anesthetized with 1% sodium pentobarbital by intraperitoneally. The brains were harvested after perfusion with 1 x PBS on ice, and the cerebellum and olfactory bulb were removed. The brains were placed in a brain matrix, frozen at −20 °C for 5–10 min, taken out and cut into 5 coronal slices with a thickness of 2 mm. The brain slices were incubated with 0.5% 2,3,5-triphenyltetrazolium chloride (TTC) in 1 x PBS for 15 min at 37 °C in thermocycler protected from light. The slices were fixed in 4% PFA and photographed by a camera. The normal brain tissue was shown in pink and the infarcted area was shown in white. Image J software analyzed and calculated the infarct volume.

Calculation formula: infarct volume = (volume of unlesioned hemisphere − infarct volume of lesioned hemisphere)/volume of unlesioned hemisphere × 100% [56].

### 4.8. Modified Neurological Severity Scores (mNSS)

Modified neurological severity scores (mNSS) is a common tool for assessing the state of rodents neurological function deficits [57]. The mNSS score includes three blocks, motor function, sensory function, and reflex loss, with a total of 18 items. Inability to complete one of these tests or absence of reflexes was counted as 1 point. The higher the score, the more serious the injury to the neurological function, 10–18 points, 5–9 points, and 1–4 points correspond to severe, moderate, and mild injury, respectively.

### 4.9. Western Blot

Rats were anesthetized with 1% sodium pentobarbital by intraperitoneally, and the brains were harvested by transcardial perfusion, lysed with RIPA buffer (Beyotime, China, P0013C), and total protein was extracted. The total protein was quantified by BCA kit, and 30 μg of total protein was loaded, the target protein was separated by 12% SDS-PAGE electrophoresis, and the protein was transferred to PVDF membrane. After protein transfer, the membrane was placed in 5% non-fat milk in 1 X TBST for blocking for 60 min, and the primary antibody was incubated at 4 °C overnight (anti-caspase-3 antibody, ab13847, 1:1000; anti-caspase-8, ab25901, 1:1000; anti-Laminin, ab11575, 1:1000; β-actin, ab179467, 1:2000). On another day, the membranes was washed 3 times with TBST for 10 min each time, the secondary antibody was incubated for 90 m at room temperature, and the membrane was washed with TBST 3 times for 10 min each time. ECL ultrasensitive luminescent solution was used for chemiluminescence.

### 4.10. Immunofluorescence

Rats were anesthetized with 1% sodium pentobarbital by intraperitoneally, and the brains were harvested by transcardial perfusion, fixed in 4% paraformaldehyde, and embedded in paraffin. Coronal sections (8 μm) of brain were cut with a microtome, dewaxed, rehydrated, and antigen retrieval was performed in citrate buffer using a microwave oven. The sections were incubated with anti-laminin antibody (ab11575, Abcam, 1:200) and anti-VEGF antibody (MA5-13182, Thermo Fisher Scientific, Rockford, 1:150) diluted with 5% skim milk, and placed at 4 °C overnight. On the second day, the wet box was taken out, placed at room temperature for 30 min, and washed three times with PBS for 10 min each time. The excess liquid was wiped off and diluted fluorescent secondary antibody (1:500) was added in the dark. Samples were incubated at room temperature for 3 h and washed 3 times with PBS for 10 min each time. Slides were mounted with DAPI glycerol and observed under a fluorescence microscope and images were captured.

### 4.11. Statistical Analyses

SPSS 21.0 software was used for comprehensive data analysis, and the data were expressed as the mean ± standard deviation (mean ± SD). The enumeration data were tested by χ^2^ test, and the measurement data were tested by *t*-test after the homogeneity of variance obeyed normal distribution, and the nonparametric test was performed if the normal distribution is violated in any subgroup. The statistical method for the comparison of scores before and after rehabilitation within the group was based on the results of the sphere test. The volume of cerebral infarction was analyzed by one-way analysis of variance, and *p* < 0.05 was considered statistically significant.

## Figures and Tables

**Figure 1 ijms-23-10508-f001:**
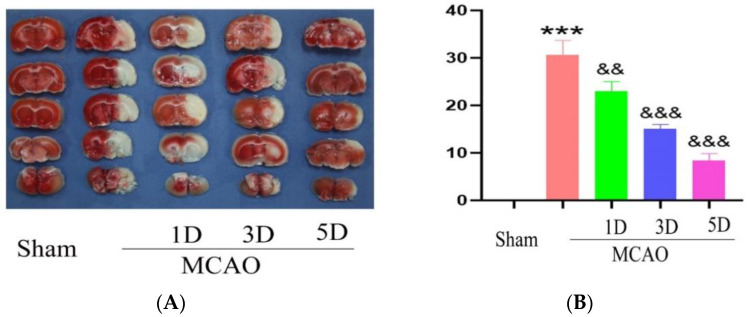
The cerebral infarction volume were detected with TTC staining method following middle cerebral artery occlusion and treadmill exercise (**A**) TTC staining method was used to detect brain sections of mouse with middle cerebral artery occlusion and treadmill exercise 1D, 3D and 5D; White is the cerebral infarction area, and red is normal area. (**B**) Statistical analysis of cerebral infarct volume in (**A**). (*** *p* < 0.001 vs. Sham; && *p* < 0.01, &&& *p* < 0.001 vs. MCAO) *n* = 5.

**Figure 2 ijms-23-10508-f002:**
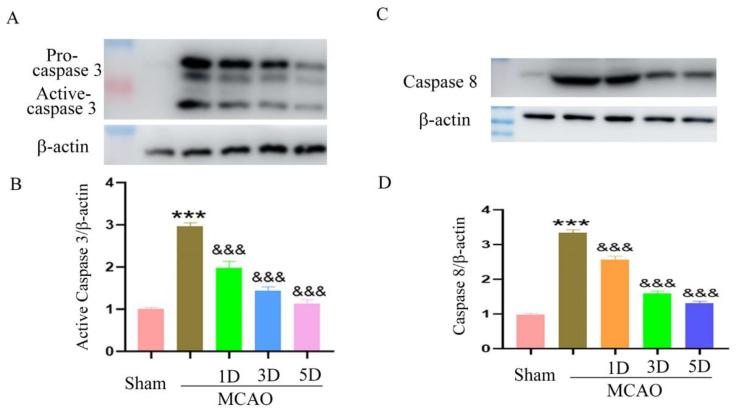
The expression of apoptosis-regulating proteins were detected by Western blot method following middle cerebral artery occlusion and treadmill exercise. (**A**) The expression level of activated forms of caspase 3 were detected with Western blot in mouse brain with middle cerebral artery occlusion and treadmill exercise 1D, 3D, and 5D; (**B**) Statistical analysis of the expression level of activated forms of caspase 3 in (**A**); (**C**): The expression level of activated forms of caspase 8 were detected with Western blot in mouse brain with middle cerebral artery occlusion and treadmill exercise 1D, 3D, and 5D; (**D**): Statistical analysis of the expression level of activated forms of caspase 8 in (**C**). (*** *p* < 0.001 vs. Sham; &&& *p* < 0.001 vs. MCAO, *n* = 3).

**Figure 3 ijms-23-10508-f003:**
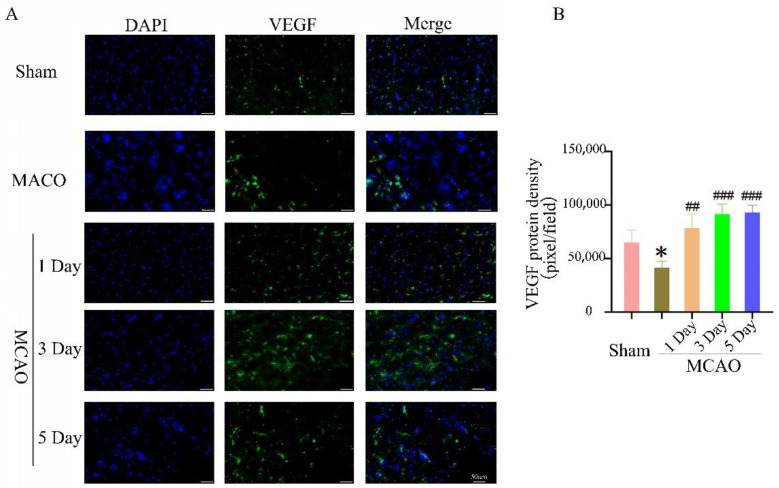
The expression of vascular endothelial growth factor were detected by immunofluorescence method follow middle cerebral artery occlusion and treadmill exercise. (**A**) The expression levels of vascular endothelial growth factor were detected with immunofluorescence in mouse brain with middle cerebral artery occlusion and treadmill exercise 1D, 3D, and 5D; bars: 50 μm; (**B**) Quantitative analysis of the expression level of a vascular endothelial growth factor in (**A**). (* *p* < 0.01 vs. Sham; ## *p* < 0.01, ### *p* < 0.001 vs. MCAO, *n* = 3).

**Figure 4 ijms-23-10508-f004:**
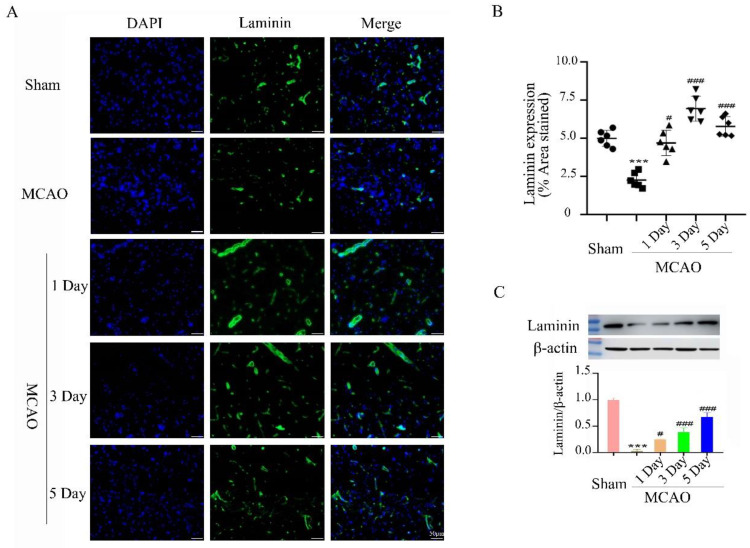
The effect of 24 h treadmill exercise after MCAO on the expression of laminin and mi crovessel density. (**A**) The laminin expression was detected by immunofluorescence method in the brain of the rats in the Sham, MCAO, 1 day, 3 days, and 5 days in treadmill exercise 24 h after MCAO groups; bars: 50 μm; (**B**) Quantification of microvessel density as shown by laminin expression (% area stained) in the brain of the rats in the Sham, MCAO, 1 day, 3 days, and 5 days in treadmill exercise 24 h after MCAO group; (**C**) The laminin expression was detected by Western blot method in the brain of the rats in the Sham, MCAO, 1 day, 3 days and 5 days in treadmill exercise 24 h after MCAO groups. (*** *p* < 0.001 vs. Sham; # *p* < 0.01, ### *p* < 0.001 vs. MCAO, *n* = 3).

**Figure 5 ijms-23-10508-f005:**
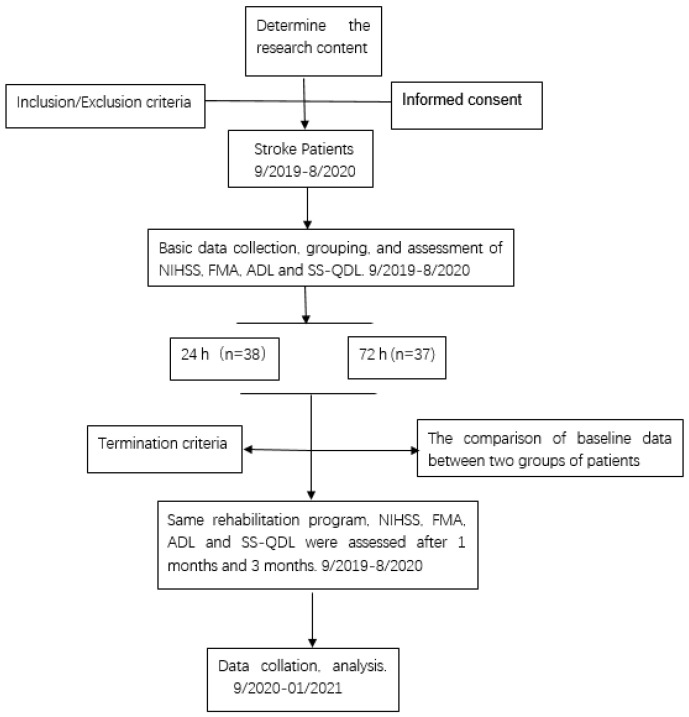
Study flow chart. The time course of this study, inclusion, exclusion, termination criteria, and study groupings are displayed graphically.

**Table 1 ijms-23-10508-t001:** The NIHSS scores of with acute ischemic stroke patients before and after rehabilitation ($\bar{x}$ ± s, *n* = 75).

Group	Before Rehabilitation	1 Month after Rehabilitation	3 Months after Rehabilitation	*F*	*p*
24 h rehabilitation	6.47 ± 1.589	3.21 ± 1.069	0.76 ± 0.852	564.881	<0.001
72 h rehabilitation	6.76 ± 1.739	5.05 ± 1.246	2.51 ± 0.989	380.673	<0.001
*t*	−0.736	−6.882	−8.216		
*p*	0.464	<0.001	<0.001		

**Table 2 ijms-23-10508-t002:** The FMA scores of with acute ischemic stroke patients before and after rehabilitation ($\bar{x}$ ± s, *n* = 75).

Group	Before Rehabilitation	1 Month after Rehabilitation	3 Months after Rehabilitation	*F*	*p*
24 h rehabilitation	15.68 ± 5.951	62.32 ± 9.286	91.18 ± 7.377	776.513	<0.001
72 h rehabilitation	16.27 ± 6.475	41.43 ± 7.065	73.22 ± 7.811	592.256	<0.001
*t*	−0.408	10.939	10.244		
*p*	0.684	<0.001	<0.001		

**Table 3 ijms-23-10508-t003:** The daily living ability scores of the acute ischemic stroke patients before and after rehabilitation ($\bar{x}$ ± s, *n* = 75).

Group	Before Rehabilitation	1 Month after Rehabilitation	3 Months after Rehabilitation	*F*	*p*
24 h rehabilitation	23.37 ± 5.196	66.95 ± 4.707	96.55 ± 2.873	543.936	<0.001
72 h rehabilitation	22.84 ± 5.036	52.95 ± 6.725	84.46 ± 6.971	316.315	<0.001
*t*	0.449	10.469	9.870		
*p*	0.655	<0.001	<0.001		

**Table 4 ijms-23-10508-t004:** Quality of life scores of the acute ischemic stroke patients before and after rehabilitation ($\bar{x}$ ± s, *n* = 75).

Group	Before Rehabilitation	1 Month after Rehabilitation	3 Months after Rehabilitation	*F*	*p*
24 h rehabilitation	114.39 ± 7.772	168.61 ± 6.323	215.95 ± 10.977	414.795	<0.001
72 h rehabilitation	113.70 ± 8.323	149.92 ± 6.264	195.78 ± 8.145	271.184	<0.001
*t*	0.372	12.855	9.015		
*p*	0.711	<0.001	<0.001		

**Table 5 ijms-23-10508-t005:** The mNSS score of rats in the 24 h treadmill exercise after MCAO ($\bar{x}$ ± s).

Days	MCAO Group	Treadmill Exercise Group in 24 h	*t*	*p*
1 d	10.22 ± 1.263	10.17 ± 1.505	0.120	0.905
3 d	8.28 ± 1.018	7.22 ± 1.478	2.496	0.018
5 d	6.00 ± 0.767	4.44 ± 0.784	6.018	<0.01

## Data Availability

Data are contained within the article or Appendix A.

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
