# Peer review of "Early Rehabilitation Exercise after Stroke Improves Neurological Recovery through Enhancing Angiogenesis in Patients and Cerebral Ischemia Rat Model"

_ijms, 2022, doi:10.3390/ijms231810508_

Round 1
Reviewer 1 Report
The study has a noble goal, that of finally analyzing the timing of post-stroke rehabilitation. Unfortunately, however, in the article I found several errors from a structural point of view:
After the introduction the results are reported, then the discussion and finally the materials and methods. This does not respect the structuring of classic papers and makes reading very difficult. In this regard, I recommend that you follow the CONSORT guidelines for RCT studies that you can find at the following link:
https://view.officeapps.live.com/op/view.aspx?src=https%3A%2F%2Fwww.equator-network.org%2Fwp-content%2Fuploads%2F2013%2F09%2FCONSORT-2010-Checklist-MS-Word.doc&wdOrigin=BROWSELINK
Referring to the CONSORT guidelines, I ask you to address all the points on the checklist as a section relating to conclusions is missing from your study, there is no flowchart with flow charts that explain the chronological order of the study and I also ask you to deepen in the introductory part the topic concerning the various types of rehabilitation used in the neuromotor field (CIMPT, Kabat, BOBATH etc).
Furthermore, the analysis part of the rats is confounding with respect to the RCT on patients. I think it is appropriate to dedicate an exclusive part to this topic in the materials and methods and then in the results, discussion and conclusions to integrate and discuss the results on patients and those on rats.
I think the study is very interesting, it deals with a very delicate subject in rehabilitation so I urge you to correct the article so that you can then approve it as it has captured my interest a lot.
Best Regards
Author Response
The study has a noble goal, that of finally analyzing the timing of post-stroke rehabilitation. Unfortunately, however, in the article I found several errors from a structural point of view:
1. After the introduction the results are reported, then the discussion and finally the materials and methods. This does not respect the structuring of classic papers and makes reading very difficult.
Response:
Thanks.
The structure format of this manuscript is the unified format given by the IJMS journal.
2. In this regard, I recommend that you follow the CONSORT guidelines for RCT studies that you can find at the following link:
https://view.officeapps.live.com/op/view.aspx?src=https%3A%2F%2Fwww.equator-network.org%2Fwp-content%2Fuploads%2F2013%2F09%2FCONSORT-2010-Checklist-MS-Word.doc&wdOrigin=BROWSELINK
Referring to the CONSORT guidelines, I ask you to address all the points on the checklist as a section relating to conclusions is missing from your study, there is no flowchart with flow charts that explain the chronological order of the study and I also ask you to deepen in the introductory part the topic concerning the various types of rehabilitation used in the neuromotor field (CIMPT, Kabat, BOBATH etc).
Response:
Thanks.
The CONSORT guidelines are highlighted in red in this revised manuscript. About CIMPT, Kabat and BOBATH etc rehabilitation types, we have discussed in introduction section in this revised manuscript (lines50-59).
3. Furthermore, the analysis part of the rats is confounding with respect to the RCT on patients. I think it is appropriate to dedicate an exclusive part to this topic in the materials and methods and then in the results, discussion and conclusions to integrate and discuss the results on patients and those on rats.
Response:Thanks.
Your comments are is highlighted and revised in the method and discussion sections, which is marked in red in in this revised manuscript.
4. I think the study is very interesting, it deals with a very delicate subject in rehabilitation so I urge you to correct the article so that you can then approve it as it has captured my interest a lot.
Response:Thank you very much for your valuable comments
Reviewer 2 Report
Please kindly find attached review comments.

Round 2
Reviewer 1 Report
Thanks for your corrections. Good job.
Best Regards
Fabio
Reviewer 2 Report
The authors have much improved the manuscript. Most of my earlier comments have been well addressed.
However, I noted some grammar/spelling errors, e.g., "is a leading causes of..." in the first sentence of the abstract. I strongly suggest the authors to find a professional native speaker for proofreading.